# Simple and Safe: Inverse Plication of the Posterior Mitral Leaflet in Everyday Mitral Valve Reconstruction with and without Concomitant Procedures

**DOI:** 10.3390/medicina59020218

**Published:** 2023-01-23

**Authors:** Roya Ostovar, Farnoosh Motazedian, Martin Hartrumpf, Filip Schröter, Johannes Maximilian Albes

**Affiliations:** Department of Cardiovascular Surgery, Heart Center Brandenburg, University Hospital Brandenburg, Medical School “Theodor Fontane”, Faculty of Health Sciences Brandenburg, 16321 Bernau bei Berlin, Germany

**Keywords:** mitral valve repair, posterior mitral leaflet, long-term survival, quadrangular resection, plication, surgical techniques

## Abstract

*Objective*: Cardiosurgical mitral valve repair (MVR) cannot be easily replaced by other interventional procedures due to the complexity of mitral valve regurgitation as well as physiologic and anatomic repair techniques. A wide variety of techniques have been adopted for proper reconstruction of posterior leaflet prolapse. We investigated the long-term results of quadrangular resections and compared them with a simplified reconstruction maneuver, the inverse plication. *Methods*: We retrospectively collected data from 1977 patients after mitral valve repair between 2004 and 2022. After considering inclusion and exclusion criteria, we analyzed data from 180 patients after MVR with and without concomitant procedures such as CABG and/or aortic valve replacement (AVR). All MVRs were performed with a semi-rigid annuloplasty ring. A total of 180 patients received quadrangular resection (QuadRes, N = 120)) or inverse plication (InvPlic, N = 60) of the posterior leaflet, among other measures. Demographic data, risk factors, procedure times, hospitalization time, early and long-term mortality as well as Kaplan–Meier survival were analyzed. *Results*: Age (65.3 vs. 63.1 years) and sex (28.8% female vs. 337.5% female) did not differ significantly, and the EuroSCORE was significantly higher in the InvPlic group (6.46 ± 2.75) than in the QuadRes group (5.68 ± 3.1). Procedural times did not differ for cardiopulmonary bypass, and were as follows: InvPlic 136 ± 44 min; QuadRes 140 ± 48 min; X-Clamp: InvPlic 91 ± 31 min; QuadRes 90 ± 32 min. Hospitalization time was slightly but insignificantly lower in the InvPlic group (15.5 days) than in the QuadRes group (16.1 days). Early mortality (5.08% vs. 8.33%) and re-do procedures (1.69% InvPlic; 6.67% QuadRes) did not differ significantly. However, long-term mortality was significantly lower in the InvPlic group (15.25% vs. 32.32%, *p* = 0.029). *Conclusions*: Among the surgical measures for the posterior leaflet, inverse plication appears to be non-inferior to quadrangular resection in unselected all-comer patients. Long-term results and absence of re-do procedures indicate very good stability. Thus, inverse plication not only allows correction of PML, but is also completely safe in the long term and can replace quadrangular resection, especially in patients where a reduction in technical challenges and procedure duration is desired.

## 1. Introduction

Posterior mitral leaflet (PML) prolapse, as a common cause of primary mitral regurgitation, is preferably treated by surgical mitral valve repair (MVR). Surgical anatomical repair cannot be simply replaced by alternative procedures because of the excellent long-term results. [1]. For over 60 years, various experimental and clinical approaches to mitral valve repair have been developed. While some techniques, such as Carpentier’s ring annuloplasty, triangular and quadrangular resection and artificial neochord implantation, are predominantly used today, other techniques, such as Kay’s annuloplasty, Wooler plasty and the Alfieri edge-to-edge repair technique, have been abandoned due to insufficient long-term results or poor anatomical physiological repair [2,3,4,5,6].

One of the cornerstones of Carpentier’s reconstruction strategies was the quadrangular resection of the posterior leaflet (PML) in case of prolapse. During our own 20 years of experience, we have developed a simplified strategy of performing an inverse plication of the PML instead of a quadrangular resection, a procedure that simplifies the process considerably, with good results and no apparent problems. In a previous study, we looked at all aspects of mitral valve repair, such as ring annuloplasties (N = 738), Wooler plastic (N = 36), NeoChord implantation (N = 168), resections on AML and PML (N = 125), patch plastic on AML and PML (N = 10), plications on both, AML and PML (N = 124) and cleft closures between 2004 and 2019. This study analyzed the evolution of mitral valve reconstruction procedures over time [7]. For the purpose of a proper comparison of quadrangular resection with inverse plication, the present study compares the outcome of “inverse plication of PML” with quadrangular resection of the PML only. We investigate the long-term stability of our simplified approach in those of our patients undergoing MVR, and compare the outcome of inverse plication and quadrangular resection on the PML.

## 2. Patients and Methods

Initially, the data of 1977 patients after mitral valve surgery with or without concomitant procedure between 2004 and 2022 were collected retrospectively. After considering inclusion and exclusion criteria, a total of 180 patients were included and analyzed in the study. The underlying mitral regurgitation pathology was assumed to be predominantly primary degenerative due to PML prolapse. However, due to the proportion of patients with concomitant myocardial revascularization, some patients could conceivably have additional secondary functional (ischemic) regurgitation according to ESC guidelines [8]. The inclusion criteria comprised patients with primary severe mitral regurgitation and presence of PML prolapse. The exclusion criteria were isolated annulus dilatation due to obvious secondary mitral regurgitation, mitral valve repair due to cardiac tumor resection and acute endocarditis. Other exclusion criteria were intraoperative chordal transfer and replacement, ultima ratio surgery, aortic dissection and rupture and patients younger than 18 years. Moreover, patients who refused verbally to participate in the study over the follow-up period were excluded from the study.

A total of 180 patients with quadrangular resection (N = 120) and patients with inverse plication (N = 60) were selected from the entire cohort for comparative analysis. The specific technical aspects are shown comparatively in Figure 1. Patients were contacted by telephone for follow-up. Long-term results up to 15 years were obtained. Survival and the need for re-do surgery were assessed during the follow-up. The data were completely anonymized after follow-up and before statistical analysis. Follow-up was performed by telephone in some cases and by outpatient visits in others.

### 2.1. Ethical Statement

The ethics vote was obtained from the responsible ethics committee (September 2020; reference E-01-20200709). In this retrospective, completely anonymized study, written informed consent was waived.

### 2.2. Statistical Analysis

Statistical analysis was performed with R^®^ 4.1.1 [9]. Descriptive statistics considered all demographic data as well as perioperative variables collected during the hospitalization and are provided as mean ± standard deviation (95% confidence interval). Numerical items were screened for normal distribution before comparison with Student’s *t*-test or the Mann–Whitney U-test. Categorical data were tested using Fisher’s exact test and the chi-square test. The presence of trends in categorical variables was analyzed with the Cochran–Armitage test for trends in proportions. In the risk factor analysis, odds ratios were calculated. Kendall’s tau was used to relate the time of hospitalization and a number of typical risk factors. Differences were considered significant when *p* was <0.05. Survival was calculated with the Kaplan–Meier method.

### 2.3. Surgical Procedure

Surgery was performed in all cases by using extracorporeal circulation and in cardioplegic arrest. Access to the mitral valve was predominantly via the left atrium. All patients received a semi-rigid, closed annuloplasty ring. The technical aspects of inverted plication are shown in Figure 1 and Figure 2 in comparison to the quadrangular resection. The Quadrangular resection was performed according to the proven steps already developed and described by Carpentier. Briefly, after determining the extent of resection in the prolapsing segment, non-elongated first-order chordae were identified on both sides adjacent to the prolapse. Then, the approximately right-angled, actually rather trapezoid resection of the corresponding segment from the free edge towards the mitral valve annulus was performed with a scalpel or scissors. In the area of the base towards the annulus, 1–2 mm remnants of the leaflet were left. The newly created vertical gap between the remaining segments was sutured with 5-0 polypropylene, starting at the upper edge, usually with single sutures. Typically, the width of the base of the defect was reduced by using more stable single or U-sutures, so that the length of the entire posterior anulus was slightly reduced. This was all carried out prior to sizing the annuloplasty ring and placing of the final sutures to anchor the ring and the actual ring implantation.

Inversion plication was performed by defining the marginal areas of the posterior leaflet adjacent to the prolapse, which were supported by non-elongated first-order chordae. An initial suture was then placed to approximate the margins so that the unaffected chordae were approximated with typical chordal spacing. The protruding leaflet tissue was moved towards the ventricle and the suture was knotted. Chordae stumps were resected to avoid later echocardiographic confusion with freshly ruptured chordae or endocarditis vegetations. Elongated chordae were left intact to serve as a “lifeline” in case of suture rupture. The valve was then tested with saline. If the coaptation was correct, the remaining gap was closed typically with two or three more U-sutures depending on the height of the posterior leaflet, which also included the area of the transition from the valve to the annulus to reduce the gap, thus improving stability of the base of the suture. Interrupted sutures were used instead of a running suture for security purposes. The valve was tested again. If sufficient coaptation was not achieved, the suture was cut and a larger exclusion area was selected. If this was not possible, the technique was discarded, and a neo-chordae implantation was performed instead. After both the quadrangular resection and the inverse plication technique, the size of the annuloplasty ring was subsequently measured and the ring implanted according to the widely used technique also described by Carpentier.

## 3. Results

### 3.1. Baseline and Comorbidities

InvPlic patients were slightly older than QuadRes patients, without a statistically significant difference (65.3 vs. 63.1 years, *p* = 0.179). No significant difference was seen in gender either, while predominantly male patients were present in both groups (71.19% vs. 62.5%, *p* = 0.327). The EuroSCORE was significantly higher in InvPlic patients (6.46 vs. 5.68, *p* = 0.039), whereas Logistic EuroSCORE and EuroSCORE II as well as urgency status did not differ significantly. The proportion of patients with coronary heart disease, peripheral arterial disease, kidney failure, chronic obstructive pulmonary disease and pulmonary hypertension did not differ significantly between the two groups. However, the proportion of patients with any of these comorbidities was slightly higher in the InvPlic group (Table 1).

### 3.2. Echocardiographic Finding

The left ventricular ejection fraction was slightly lower in the InvPlic group, and tricuspid annular plane systolic extension (TAPSE) was mildly worse without a significant difference (56% vs. 58%, *p* = 0.303; 15.54 vs. 18.35 mm, *p* = 0.33). Pulmonary arterial pressure was similar in both groups. No significant difference could be found between the groups regarding left atrial diameter, diastolic and systolic left ventricular diameters, effective regurgitant orifice area, vena contracta and regurgitation fraction (Table 1).

### 3.3. Surgical Procedure

The percentage of isolated repairs did not differ significantly between the two groups. Concomitant CABG was compared equally frequently in both groups. InvPlic patients underwent concomitant aortic valve replacement (AVR) more frequently than the QuadRes group. Concomitant CABG AND AVR was not performed in the InvPlic group, but was performed in seven patients in the QuadRes group. However, this difference was not significant. Cardiopulmonary bypass time and X-clamp time did not show significant differences between the InvPlic and QuadRes groups (136.35 vs. 140.48 min, *p* = 0.669 and 90.86 vs. 89.5 min, *p* = 0.807, respectively) (Table 2).

### 3.4. Outcome and Mortality

Hospitalization time was slightly shorter in the InvPlic group without reaching statistical significance (15.5 vs. 16.1 days, *p* = 0.708). Discharge status did not differ between the groups. Re-do procedures were more frequent in the QuadRes group over the long-term follow-up, but without statistical significance (6.67% vs. 1.69%, *p* = 0.275). Concerning postoperative complications, no significant difference was found between the groups in terms of stroke, postoperative hemorrhage and pericardial tamponade, low cardiac output syndrome, pleural effusion, pneumonia, pneumothorax, wound healing disorders, delirium, urinary tract infection and renal insufficiency (Table 2).

There was no difference in early mortality. Long-term all-cause mortality was significantly higher in the QuadRes group than in the InvPlic group (early mortality 8.33% vs. 5.08%, *p* = 0.55; long-term mortality 32.32% vs. 15.25%, *p* = 0.029, respectively). However, as InvPlic was started later in this retrospective analysis, in contrast to QuadRes, results for more than 10 years are not yet available (Figure 3).

## 4. Discussion

Mitral valve repairs are still performed in considerable numbers despite the increasing number of MitraClip procedures, in contrast to AVR, which is increasingly being replaced by TAVI [5]. The reason for this is the complexity of the mitral valve. A true anatomical repair is not possible with the MitraClip because the edge-to-edge connection suffers from high and non-physiological tension forces that eventually lead to tissue overload and thus failure of the procedure [10]. Therefore, surgical repair is often advocated. Due to demographic change, the indication has shifted from isolated mitral valve surgery to concomitant procedures. Especially in these situations, a permanent surgical repair is necessary and desirable to achieve good procedural quality and thus long-term stability without re-do interventions. To achieve this goal, an annuloplasty ring is required to increase the coaptation area of both leaflets and remodel and stabilize the annulus. There is increasing evidence of the benefits of the ring. Rigid or semi-rigid rings seem to stabilize the annulus considerably more effectively than highly flexible ones [11,12]; thus, the patients with flexible rings were not included in our study. In contrast, many different techniques are used to repair the leaflet and leaflet-retaining apparatus. These include classical resection maneuvers or the use of artificial chordae to reshape the leaflet and achieve proper coaptation [13,14,15,16,17]. Especially in older patients with significant risk factors and in patients who require concomitant procedures, surgeons are looking for procedures that facilitate and speed up the whole process. The Alfieri stitch, the surgical predecessor of the MitraClip, was one such technique. Nowadays, however, it is seldom performed surgically although being very fast [10,11,12,13,14,15,16,17,18]. During our 20 years of experience, we came across a simplification of the classic quadrangular resection maneuver by performing a reverse plication instead of resection or a neo-chordae implantation. This maneuver was initially only intended to check whether proper coaptation could be achieved with the subsequent resection of the desired segment. However, it proved to be so simple and quick that we began to dispense with the resection step. In this way, the total procedure time could be limited or used for other demanding tasks, such as difficult bypass anastomoses or aortic valve implantations. Over time, it became clinically apparent that patients in whom this technique was initially used primarily for reasons of speed later did not present with inadequate outcomes, such as residual significant regurgitation, nor were prone to recurrences requiring reinterventions. To verify this promising first-hand clinical observation, this study was conducted. The two cohorts were very similar regarding baseline, risk profile and comorbidities, and the echocardiographic parameters were also similar, so we did not run the risk of heterogeneity in the comparison. The results do not show any significant difference in mortality, outcome or re-do intervention. In direct comparison with patients who underwent the classical resection maneuver, we were able to clearly demonstrate the non-inferiority of this technique, even in the long term. We even assume another safety advantage of this technique. In the event of re-dehiscence due to rupture of one or more sutures, no serious problems endangering the patient are conceivable apart from recurrence of prolapse and resulting mitral regurgitation, quite unlike rupture after quadrangular resection, which would inevitably lead to severe mitral regurgitation. However, this could not be proven, as such problems did not occur in our QuadRes patients. Furthermore, it can even be speculated that this technique is safer than the neo-chordae approach, which can occasionally rupture further down the line, requiring reintervention [6,19].

### Limitations

This study is retrospective in nature and thus naturally subject to the typical limitations. The procedures were performed at the discretion of the respective surgeon and were not prospectively randomly assigned. In the early period from 2004 to 2006, intraoperative echocardiography was not considered mandatory and specific documentation was missing. Furthermore, surgery from these years often did not elucidate the presence or absence of intraoperative echocardiography.

## 5. Conclusions

Inverse plication can be used instead of quadrangular resection in patients with posterior leaflet prolapse. It is much less time-consuming, less demanding, even simple and completely stable. This technique is not suggested for replacing classical resection or the use of artificial chordae, but it can be recommended judiciously for those cases in which the surgeon wishes to steer expeditiously through what is likely to be a lengthy procedure.

## Figures and Tables

**Figure 1 medicina-59-00218-f001:**
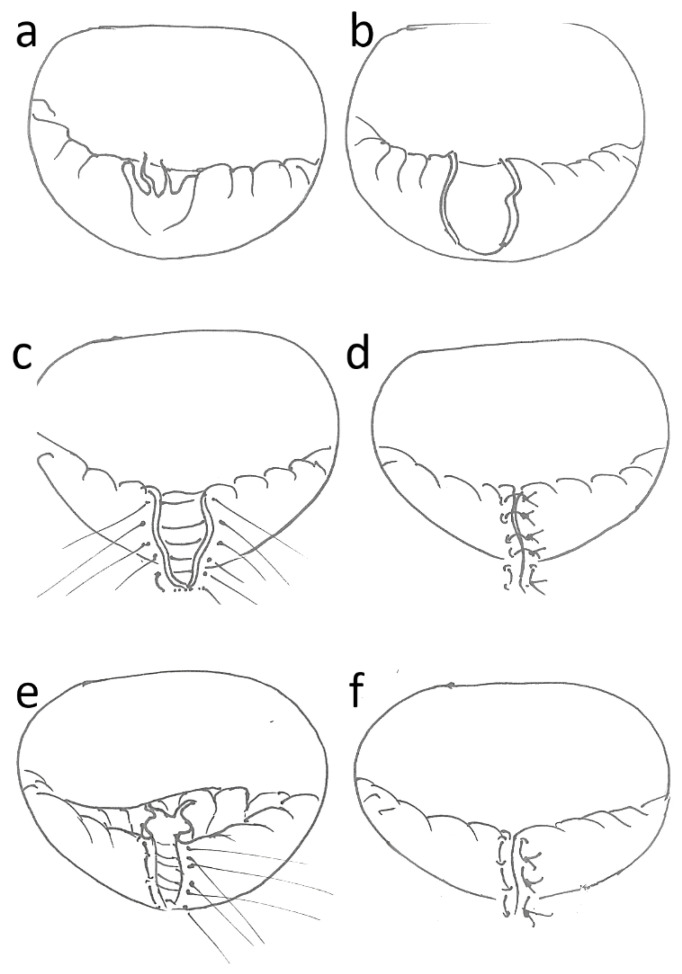
PML repair: (**a**): PML prolapse with severed chordae; (**b**): quadrangular resection; (**c**): suture placement (5-0 polypropylene); (**d**): result after quadrangular resection; (**e**): inversion of prolapsed leaflet and suture placement (U-stiches, 5-0 polypropylene); (**f**): results after inverse plication.

**Figure 2 medicina-59-00218-f002:**
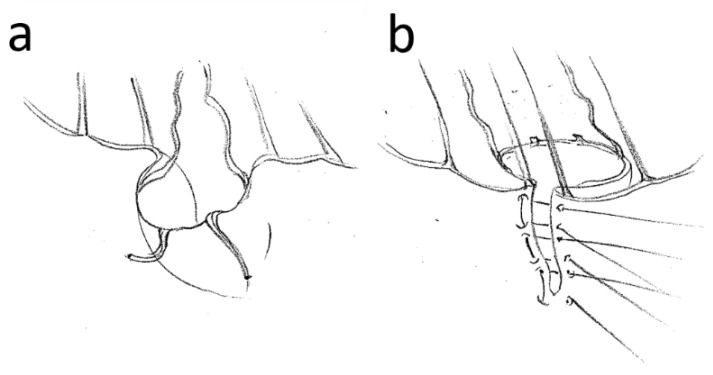
PML repair with the transverse plication technique. Detailed view of the protruding area (**a**) and the corrective measures, including the definition of the leaflet areas with the chordae still intact, the reversal of the prolapse and the closure suture technique (**b**). Note the resected chordae stumps and the extended chordae, which were left intact.

**Figure 3 medicina-59-00218-f003:**
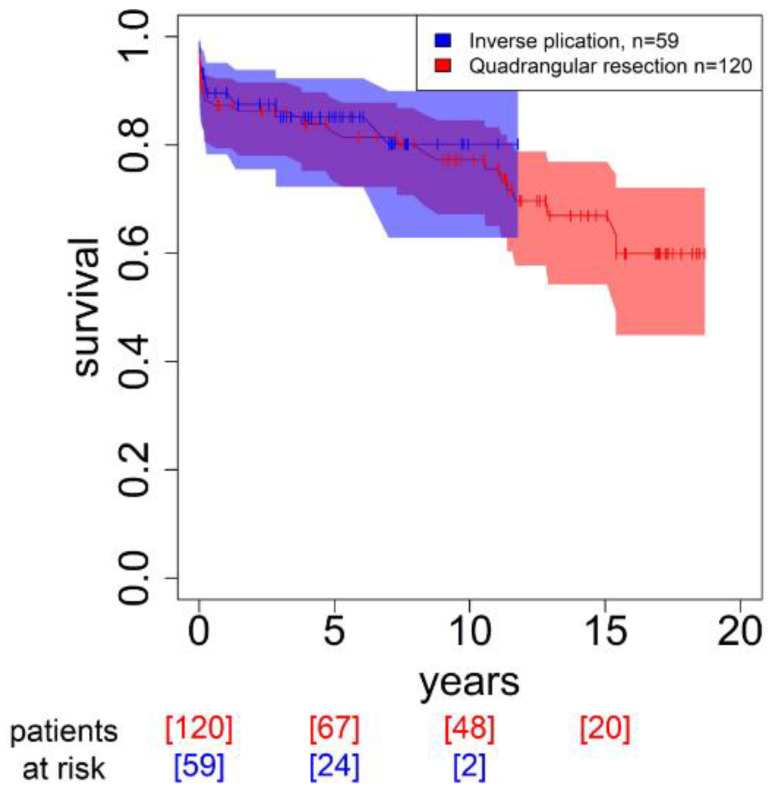
Kaplan–Meier survival curve.

**Table 1 medicina-59-00218-t001:** Baseline.

	InvPlic (n = 60)	QuadRes (n = 120)	*p*
Age (years)	65.27 ± 13.21	63.1 ± 12.52	0.179
Female sex (%)	28.81	37.5	0.327
EuroSCORE	6.46 ± 2.75	5.68 ± 3.11	0.039
Logistic EuroSCORE (%)	8.71 ± 9.55	9.05 ± 12.82	0.225
EuroSCORE II	4.2 ± 5.24	1.96 ± 1.92	0.142
Coronary heart disease (%)	40.54%	31.25%	0.509
Peripheral arterial disease	2.7%	0%	0.435
Kidney failure	13.51%	12.5%	1
COPD	5.41%	4.35%	1
Pulmonary hypertension	16.22%	10.87%	0.698
Urgency status			0.438
Elective (%)	49.15%	59.17%	0.268
Urgent (%)	50.85%	37.5%	0.124
Emergent (%)	0%	3.33%	0.304
Echocardiographic finding			
Left ventricular ejection fraction (%)	56.11 ± 8.95	57.9 ± 11.75	0.303
TAPSE (mm)	15.54 ± 9.55	18.35 ± 5.06	0.33
Systolic left ventricular diameter (mm)	33.58 ± 4.35	35.99 ± 9.17	0.663
Diastolic left ventricular diameter (mm)	49.98 ± 9.25	51.78 ± 11.71	0.302
Left atrial diameter (mm)	41.77 ± 10.55	43.15 ± 10.84	0.861
Pulmonary arterial pressure (mmHg)	28.86 ± 9.55	29.05 ± 14.19	0.814
Vena contracta (mm)	9.1 ± 2.18	9.2 ± 2.7	0.928
EROA (cm^2^)	41.25 ± 12.68	32.33 ± 15.76	0.261
Regurgitation fraction (%)	58 ± 5.87	54.33 ± 6.95	0.368

ES: EuroSCORE; log. ES: logistical EuroSCORE; ESII: EuroSCORE II; COPD: chronic obstructive pulmonary disease; TAPSE: tricuspid annular plane systolic extension; EROA: effective regurgitant orifice area; QuadRes: quadrangular resection; InvPlic: inverse plication.

**Table 2 medicina-59-00218-t002:** Outcome and mortality.

	InvPlic (n = 60)	QuadRes (n = 120)	*p*
CPB time (min)	136.35 ± 43.89	140.48 ± 48.15	0.669
X-Clamp time (min)	90.86 ± 30.75	89.5 ± 31.86	0.807
Annuloplasty ring (%)	100	100	1.000
Hospitalization time (day)	15.53 ± 8.7	16.1 ± 11.81	0.708
Re-do procedure (%)	5.08%	0% (8)	0.035
Discharge status			0.953
Discharged (%)	66.1%	56.67	0.295
Rehabilitation (%)	10.17%	13.33%	0.716
Transferred (%)	16.95%	21.67%	0.588
Isolated MV repair (%)	40.68%	60.83%	0.017
With CABG (%)	15.25%	24.17%	0.239
With AVR (%)	16.95%	9.17%	0.203
Postoperative complications			
Stroke	1.67% (1)	0%	0.863
Low cardiac output syndrome	0%	0.83% (1)	1
Postoperative hemorrhage	2.22% (1)	2.04% (1)	1
Pneumonia	5% (3)	0.83% (1)	0.313
Wound healing disorders	0%	2,25% (2)	0.244
Early mortality	5%	8.33 % (10)	0.55
Long-term mortality	15.25 (7)	32.32 (32)	0.029

CPB: cardiopulmonary bypass; X-clamp: cross-clamping; CABG: aorto-coronary bypass graft; AVR: aortic valve replacement; SIRS: systemic inflammatory response syndrome; QuadRes: quadrangular resection; InvPlic: inverse plication.

## Data Availability

Data is unavailable due to ethical restrictions.

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
