# Peer review of "Simple and Safe: Inverse Plication of the Posterior Mitral Leaflet in Everyday Mitral Valve Reconstruction with and without Concomitant Procedures"

_medicina, 2023, doi:10.3390/medicina59020218_

Round 1

Reviewer 1 Report (Previous Reviewer 2)

Dear author, 

Thank you again for your revised paper. 

You clarified most of the points. In the new article is more simple to appreciate the aim of the study and the surgical procedure. 

The only remaining concern is about the phrases " The Alfieri stitch, the surgical predecessor of the MitraClip, was one such facilitated technique. Nowadays , however, it is almost obsolete as it offers no real advantage over anatomical repair" (page 8, lines 214-216). Actually, in the refrences (16 -17) there's no mention of inferiority of Alfieri stitch in comparison with other surgical techniques. In literature, instead, we can found articles showing optimal long term results (see "Rubino AS, Mignosa C, Di Bartolo M, Cavallaro A, Castorina S, Gentile M, Patanè L. Follow-up clinico ed ecocardiografico a lungo termine dopo riparazione valvolare mitralica con tecnica “edge-to-edge” [Long-term clinical and echocardiographic follow-up of the edge-to-edge technique for surgical mitral valve repair]. G Ital Cardiol (Rome). 2020 Mar;21(3):209-215. Italian. doi: 10.1714/3306.32769. PMID: 32100733." or "De Bonis M, Lapenna E, Taramasso M, La Canna G, Buzzatti N, Pappalardo F, Alfieri O. Very long-term durability of the edge-to-edge repair for isolated anterior mitral leaflet prolapse: up to 21 years of clinical and echocardiographic results. J Thorac Cardiovasc Surg. 2014 Nov;148(5):2027-32. doi: 10.1016/j.jtcvs.2014.03.041. Epub 2014 Mar 27. PMID: 24755329") as well as its use in other concomitant cardiac procedures (see for example "Mihos CG, Escolar E, Fernandez R, Nappi F. A systematic review and pooled analysis of septal myectomy and edge-to-edge mitral valve repair in obstructive hypertrophic cardiomyopathy. Rev Cardiovasc Med. 2021 

Good luck for your article and thank you again for submitting it. 

Author Response

Reviewer 1

Comments and Suggestions for Authors

Dear author, 

Thank you again for your revised paper. 

You clarified most of the points. In the new article is more simple to appreciate the aim of the study and the surgical procedure. 

The only remaining concern is about the phrases " The Alfieri stitch, the surgical predecessor of the MitraClip, was one such facilitated technique. Nowadays , however, it is almost obsolete as it offers no real advantage over anatomical repair" (page 8, lines 214-216). Actually, in the refrences (16 -17) there's no mention of inferiority of Alfieri stitch in comparison with other surgical techniques. In literature, instead, we can found articles showing optimal long term results (see "Rubino AS, Mignosa C, Di Bartolo M, Cavallaro A, Castorina S, Gentile M, Patanè L. Follow-up clinico ed ecocardiografico a lungo termine dopo riparazione valvolare mitralica con tecnica “edge-to-edge” [Long-term clinical and echocardiographic follow-up of the edge-to-edge technique for surgical mitral valve repair]. G Ital Cardiol (Rome). 2020 Mar;21(3):209-215. Italian. doi: 10.1714/3306.32769. PMID: 32100733." or "De Bonis M, Lapenna E, Taramasso M, La Canna G, Buzzatti N, Pappalardo F, Alfieri O. Very long-term durability of the edge-to-edge repair for isolated anterior mitral leaflet prolapse: up to 21 years of clinical and echocardiographic results. J Thorac Cardiovasc Surg. 2014 Nov;148(5):2027-32. doi: 10.1016/j.jtcvs.2014.03.041. Epub 2014 Mar 27. PMID: 24755329") as well as its use in other concomitant cardiac procedures (see for example "Mihos CG, Escolar E, Fernandez R, Nappi F. A systematic review and pooled analysis of septal myectomy and edge-to-edge mitral valve repair in obstructive hypertrophic cardiomyopathy. Rev Cardiovasc Med. 2021 

Good luck for your article and thank you again for submitting it. 

  • Answer: Thank you for reading and reviewing the paper. From clinical experience we know that after edge-to-edge often a residual mitral regurgitation as well as an additional mitral stenosis are present. Furthermore, after edge-to-edge, one large regurgitant jet may turn into 2 small jets and thus the residual regurgitation is often underestimated. Nonetheless, we do not find studies sufficiently reporting this. You are completely right. In fact, in most studies Alfieri technique is reported with good long-term results. However, clinical use has gradually declined and anatomical repair has been favoured instead. For this reason, we have changed the staement in the revised manuscript.

Reviewer 2 Report (Previous Reviewer 3)

Thank you for making the suggested changes.

Author Response

Reviewer 2

Comments and Suggestions for Authors

Thank you for making the suggested changes.

  • Answer: Thank you for the review of the paper, which contributed to the qualitative improvement of the study.

This manuscript is a resubmission of an earlier submission. The following is a list of the peer review reports and author responses from that submission.

Round 1

Reviewer 1 Report

None

Author Response

Thank you

Reviewer 2 Report

Thank you for submitting this article to Medicina Journal. I was pleased to receive it as a reviewer. This is a topic that could arise interest. Even if the two groups are not fairly even in number and the “InvPlic” population is not that much, it can certainly be considered a compelling subject. The main point in favor is the focus on the inverse plication of posterior leaflet, not so much represented in literature.  

In only have the questions below that should be addressed before the publication. 

First, is it a comparison between inverse posterior leaflet plication and quadrangular resection or the techniques compared are inverse posterior leaflet plication and every type of posterior leaflet resection? That is not so clear in the abstract. 

Then, I cannot find a precise description of the proposed procedure. Can you explain if it’s better to start from the top or from the base of the leaflet? Also, it could be useful knowing what type of thread is better to use and if you suggest continuous suture or single stiches.

I’ve got a concern about the affirmation that the anatomical reconstruction could properly be consider the reason of the increased durability of surgical mitral valve repair compare with percutaneous procedures. To the best of our knowledge, edge-to-edge had shown satisfactory long-term results. Don’t you think that the reason of recurrence of mitral regurgitation using Mitra Clip could be addressed to the absence of concomitant annuloplasty?

In your baseline data, I cannot find the STS score. Do you have it in your database?

I cannot find grammatical or syntax errors. 

Good luck for your paper and thak you again for submitting this article to Medicina Journal. 

Author Response

Reviewer 2

Thank you for submitting this article to Medicina Journal. I was pleased to receive it as a reviewer. This is a topic that could arise interest. Even if the two groups are not fairly even in number and the “InvPlic” population is not that much, it can certainly be considered a compelling subject. The main point in favor is the focus on the inverse plication of posterior leaflet, not so much represented in literature.  In only have the questions below that should be addressed before the publication. 

Answer: Dear Reviewer, Thank you for your valuable comments.

First, is it a comparison between inverse posterior leaflet plication and quadrangular resection or the techniques compared are inverse posterior leaflet plication and every type of posterior leaflet resection? That is not so clear in the abstract. 

Answer: The inverse plication technique was compared with posterior leaflet quadrangular resections. It is specified in the abstract and in the methods. Abstract: page 2, lines 9 and 10. Methods, section surgical procedure: highlighted in yellow.

Then, I cannot find a precise description of the proposed procedure. Can you explain if it’s better to start from the top or from the base of the leaflet? Also, it could be useful knowing what type of thread is better to use and if you suggest continuous suture or single stiches.

Answer: The technique is described under “surgical procedure, now highlighted in yellow” and also illustrated. Indicated there is starting at the top of the leaflet in order to achieve proper edge alignment first. Thereafter, the alignment underwent a first test with saline. Then the remaining gap was closed with interrupted U-sutures with 5-0 Prolene as already indicated in the respective section. Single sutures appear to be safer than continuous sutures in our humble opinion. If the continuous suture fails, everything fails, if one of the interrupted sutures fail, only a minor gap will reappear. We have added a statement regarding security using interrupted sutures (Surgical method: highlighted in green).

I’ve got a concern about the affirmation that the anatomical reconstruction could properly be consider the reason of the increased durability of surgical mitral valve repair compare with percutaneous procedures. To the best of our knowledge, edge-to-edge had shown satisfactory long-term results. Don’t you think that the reason of recurrence of mitral regurgitation using Mitra Clip could be addressed to the absence of concomitant annuloplasty?

Answer: Thank you very much for your valuable comment. Yes, indeed mitral annuloplasty is considered mandatory in all surgical reconstruction manoeuvres, even if leaflet repair was the main step, because it avoids subsequent dilatation of the annulus. So, this is one of the main advantages of surgical repair over pure edge-to-edge techniques. Furthermore, in one of our previous papers on mitral valve repair, we encountered the problem of clip failure after MitraClip edge-to-edge repair, suggesting that the edge-to-edge manoeuvre, whether surgical or interventional, is indeed not as stable as an anatomical reconstruction, probably because of the non-physiological tension forces that constantly tear at the connected edges and eventually overload the tissue. We added a respective paragraph in the discussion (page 10, lines 4-6, highlighted in green). The relevant literature was already cited in our manuscript. (Lit. 17). The relevance of annuloplasty was also mentioned in discussion (page 10, lines 10-13, highlighted in yellow).

In your baseline data, I cannot find the STS score. Do you have it in your database?

Answer: No, of course STS score takes into account some additional parameters, such as liver function, which are important for risk assessment, on the other hand is a bit more time consuming. we use EuroSCCORE in Germany. This score is well validated and is widely used in Germany and Europe.

I cannot find grammatical or syntax errors. Good luck for your paper and thak you again for submitting this article to Medicina Journal. 

Answer: Thank you again.

Reviewer 3 Report

The authors retrospectively investigated the long-term results of the quadrangular resection compared with a simplified reconstruction maneuver, the inverse plication, in patients with posterior mitral valve prolapse and severe mitral regurgitation. Inverse plication appears to be non-inferior to quadrangular resection, it is less time-consuming and less demanding.

The manuscript is well written and the topic is of importance. I have a few comments:

Page 2, lines 78-80

This exclusion criterion is repeated in lines 73-75:

“Patients who verbally refused to participate in the study during follow-up or for whom they or their relatives were unable to give informed  consent were excluded” 

Page 2, lines 64-66

The authors wrote “an additional secondary functional (ischemic) origin according to current ESC guidelines was imaginable in view of the significant proportion of patients undergoing additional CABG surgery”

What percentage of patients in each group had mitral regurgitation of mixed etiology (primary and functional)?

Page 5, Table 1

The p-value for variable COPD is missing

Page 6, Table 2

The units for hospitalization time are incorrect, it seems to me that it should be days

Page 6, first paragraph

I wonder if perioperative transesophageal echocardiograms were performed to evaluate the immediate surgical result. 

If so, in what percentage of patients was it necessary to re-operate according to the findings of the perioperative echocardiogram?

Page 7, Figure 3

The figure caption is missing.

Page 8, line 220

The article in reference 17 does not seem to have any relationship with what is written in the text.

Author Response

Reviewer 3

The authors retrospectively investigated the long-term results of the quadrangular resection compared with a simplified reconstruction maneuver, the inverse plication, in patients with posterior mitral valve prolapse and severe mitral regurgitation. Inverse plication appears to be non-inferior to quadrangular resection, it is less time-consuming and less demanding. The manuscript is well written and the topic is of importance. I have a few comments:

 Answer: Thank you for your valuable comments.

Page 2, lines 78-80

This exclusion criterion is repeated in lines 73-75: “Patients who verbally refused to participate in the study during follow-up or for whom they or their relatives were unable to give informed  consent were excluded” 

 Answer: Thank you for detecting this. We have erased the duplicate.

Page 2, lines 64-66

The authors wrote “an additional secondary functional (ischemic) origin according to current ESC guidelines was imaginable in view of the significant proportion of patients undergoing additional CABG surgery”

What percentage of patients in each group had mitral regurgitation of mixed etiology (primary and functional)? 

 Answer: Due to the retrospective nature of the study, the underlying pathology could not be comprehensively assessed in terms of true primary or secondary mitral valve regurgitation. As this work deals exclusively with the entity "posterior valve prolapse", we considered this as the primary aetiology. However, in those patients who presented with additional coronary artery disease, a mixed aetiology was assumed. Of course, this is also only an assumption, since even in prospective studies an additional secondary cause often cannot be compellingly and unequivocally proven. Regarding the proportion of mixed aetiology, we reported in table 1 that 40.54% of the inverted plication cohort and 31.25% of the quadrangular resection cohort had concomitant coronary artery disease and in Table 2, 17.5% of the inverted plication cohort and 20.75% of the quadrangular resection cohort underwent concomitant aorto-coronary bypass surgery.

Page 5, Table 1

The p-value for variable COPD is missing.

 Answer: Yes, indeed, thank you, we added the P-value in table 1.

Page 6, Table 2

The units for hospitalization time are incorrect, it seems to me that it should be days.

 Answer: Yes, indeed! Thank you. That is correct. We added the unit in the table 2.

Page 6, first paragraph

I wonder if perioperative transesophageal echocardiograms were performed to evaluate the immediate surgical result.

 Answer: Yes, intraoperative echocardiography was performed in the vast majority of all patients to immediately assess the outcome. However, in the early years (2004-2006), not all patients received mandatory intraoperative echocardiography. Some were therefore missing. Surgeons' files and protocols from these years often do not clearly indicate whether intraoperative TEE was performed or not. This is not ideal, but unfortunately, we had to live with this in this retrospective study. We have added a corresponding explanation in the limitations. Page 11, lines 10-13, Highlighted in green.

If so, in what percentage of patients was it necessary to re-operate according to the findings of the perioperative echocardiogram?

 Answer: Only one patient of the inverted plication group had a redo whereas 8 of the quadrangular resection patients underwent one. Fortunately, none of them were direct intraoperative redo’s (table 2, highlighted in yellow).

Page 7, Figure 3

The figure caption is missing.

 Answer: Yes, thank you. We added it.

Page 8, line 220

The article in reference 17 does not seem to have any relationship with what is written in the text.

 Answer: In this article we analyzed MitraClip failures and how to deal with them. This citation serves the purpose of addressing the issue of the non-anatomical repair resulting in non-physiological tension forces that constantly tear at the connected edges and eventually overload the tissue. We added a respective paragraph in the discussion (page 10, lines 4-6, highlighted in green).